# Tipping Points and Changes in Australian Climate and Extremes

**Jorgen S. Frederiksen [1,2,*] and Stacey L. Osbrough [1,2]**

1   CSIRO Oceans and Atmosphere, Aspendale, Melbourne 3195, Australia; stacey.osbrough@csiro.au
2   School of Earth, Atmosphere and Environment, Monash University, Clayton, Melbourne 3800, Australia
*   Correspondence: jorgen.frederiksen@csiro.au

**Abstract:** Systematic changes, since the beginning of the 20th century, in average and extreme Australian rainfall and temperatures indicate that Southern Australian climate has undergone regime transitions into a drier and warmer state. South-west Western Australia (SWWA) experienced the most dramatic drying trend with average streamflow into Perth dams, in the last decade, just 20% of that before the 1960s and extreme, decile 10, rainfall reduced to near zero. In south-eastern Australia (SEA) systematic decreases in average and extreme cool season rainfall became evident in the late 1990s with a halving of the area experiencing average decile 10 rainfall in the early 21st century compared with that for the 20th century. The shift in annual surface temperatures over SWWA and SEA, and indeed for Australia as a whole, has occurred primarily over the last 20 years with the percentage area experiencing extreme maximum temperatures in decile 10 increasing to an average of more than 45% since the start of the 21st century compared with less than 3% for the 20th century mean. Average maximum temperatures have also increased by circa 1 °C for SWWA and SEA over the last 20 years. The climate changes in rainfall an d temperatures are associated with atmospheric circulation shifts.

**Keywords:** Australian climate; extremes; rainfall; temperatures; tipping points; regime transitions; circulation; storms; climate change

## 1. Introduction

Over the last seventy years, since the middle of the 20th century, aspects of Australian climate, particularly rainfall and temperatures, have undergone significant changes [1–5]. The notable rainfall deficits in southern Australia have been linked to declines in extratropical storminess and the intensity of explosive storms [1,6–14]. Some of those changes have been quasi-cyclical due, for example, to variability associated with the El Niño-Southern Oscillation, the Interdecadal Pacific Oscillation, the Walker and Hadley Circulations, or the Indian Ocean Dipole [1,15–22]. On the other hand, there is also compelling evidence for systematic climate shifts in both hemispheres due to global warming [1–3,8,12,23–29].

Frederiksen, et al. [29], Figure 7, consider the systematic or secular trends in Southern Hemisphere winter baroclinicity over the second half of the 20th century based on reanalysis data and the interannual and decadal variability about the trend. Frederiksen, et al. [12] made similar determinations for each season and studied the roles of externally forced and internal covariability of rainfall and baroclinicity in a suite of 12 comprehensive coupled ocean atmosphere climate models. These models were chosen on their ability to produce similar trends in baroclinicity over the second half of the 20th century as found in reanalysis data. They also examined these model trends for the second half of the 21st century under conditions of strong radiative forcing by increasing carbon dioxide with no stabilization and, as well, with stabilization. In this study, and in a similar study by Frederiksen and Grainger [24] on the covariability of rainfall and 500 hPa geopotential height, it was concluded that the secular trends, in rainfall and circulation over the second half of the 20th century by the ensembles of skilful models, which are similar to those

from reanalyses, are closely reproduced by the externally forced modes of covariability. Moreover, the continuing similar secular trends into the 21st century are conditional on the continuing increasing trend in equivalent carbon dioxide without stabilization. The attribution study of Franzke, et al. [23] led to their conclusion that anthropogenic carbon dioxide is the dominant cause of secular changes in the Southern Hemisphere circulation in recent decades with a lesser contribution from stratospheric ozone depletion. They also noted the consistency with the model-based study of Freitas, et al. [27].

Our particular interest in this article is whether the changes that have occurred in Australian climate and climate extremes over the last seventy years are indicative of regime transitions in a noisy environment. There has been a long history of studies examining the possibility of regime transitions in various aspects of the climate system. The early simple energy balance models (EBMs) of the earth's climate [30–35] exhibited thermodynamical regime transitions in the mean temperature between several states as the order parameter, the solar constant, is varied. Indeed, as shown in Figures 1 and 3 of Frederiksen [35], the number of stable states and the number of bifurcation points (or critical or tipping points) may vary depending on the form of the thermodynamical functions, such as the effective albedo, and lead to the possibility of closely spaced tipping points.

Charney and Devore [36] and Wiin-Nielsen [37] studied low order dynamical models of the atmospheric circulation and found multiple equilibrium states dominated by either strong zonal flow and weak wave structure or weak zonal flow and strong wave structure that they interpreted as a blocking state. Charney and Devore [36] found that regime transitions between the zonal and blocking states occurred as the order parameter, the height of the topography, varied through the bifurcation point. Similar regime transitions were also found in baroclinic models by Charney and Straus [38]. Frederiksen and Frederiksen [39] reviewed subsequent developments in the theory of multiple equilibria and the role of topographic instability in regime transitions.

Frederiksen [40,41] examined regime transitions of inviscid barotropic and baroclinic zonal flows over topography in high dimensional systems using methods of equilibrium statistical mechanics. The critical points for barotropic flow and critical lines and triple points for baroclinic flows were determined and the similarities and differences with magnetic phase transitions [42,43] were examined. Zidikheri, et al. [44] studied the interaction of barotropic zonal flows with topography in high resolution forced dissipative numerical simulations and established the phase diagram (their Figure 2) for regime transitions. They found hysteresis effects in transitions between strong and weak zonal flow states with qualitative similarities to those for magnetic phase transitions (e.g., Figure 3 of Saghayezhian, et al. [45] and references therein). The regime transitions between strong zonal states and blocking found in simple models have also been found in comprehensive weather prediction models (e.g., Frederiksen, et al. [46]) and associated with observed climate shifts by O'Kane, et al. [26].

Further developments in the role of regime transitions and tipping points in various aspects of the climate system, including under global warming, have been considered by Franzke, et al. [23], Freitas, et al. [27], Dijkstra [47], Kypke, et al. [48], Lenton, et al. [49], Yan, et al. [50], Fabiano, et al. [51], Jones and Ricketts [52] and Australian Academy of Science [53]. It is clear from all the studies mentioned in this Introduction that there are dynamical and thermodynamical processes of the climate system that can result in regime transitions. However, the methodologies for analysing components and simplifications of the climate system are not easily applied to the full system given its complex equations and interactions over vast scales. This is clearly the case for the analytical and semi-analytical bifurcation methods, including singularity theory [54], for analysing low order systems [55] and for the equilibrium statistical mechanics methods [40,41]. Renormalization group methods [56–58] and renormalized perturbation theory [56,59] are more generally applicable to the statistical dynamics of phase transitions of high dimensional systems. However, they are most suited to systems described by a few mathematically elegant equations such as the Navier–Stokes equations or quasigeostrophic equations [59]. The

complex equations, some of which include discontinuous processes such as convection, and vastly different time scales of interactions, of the climate system again make these approaches unfeasible for computational as well as theoretical reasons. In this study we therefore take an approach based on the general characteristics of phase transitions which involve a discontinuity in the dependent variable (first order phase transition) or its derivative (second order phase transition) as the order parameter transits through a critical point [45,60].

The paper is structured as follows. Section 2 outlines the data and sources as well as the methodology for their analysis used in this study. The mean and extreme rainfall, streamflow into Perth dams, the mean and extreme surface temperature data sets, and the reanalysis data determining atmospheric flow fields, are described in this Section. In Section 2, the calculation of decile data including for extreme rainfall and surface temperatures is also described as is the method for establishing the critical times of the changes in the trends in the data. Section 3 examines changes in SWWA mean and extreme rainfall and streamflow since the beginning of the 20th century and relates the changes to those of the atmospheric circulation in the surrounding regions. There, systematic shifts in these variables and their trends or gradients over different time periods are examined and are related to regime transitions. In Section 4, a corresponding analysis is performed for mean and extreme rainfall for SEA and in Section 5 results for Northern Australia are presented. The changing nature of SWWA average and extreme maximum surface temperatures are examined in Section 6 and the shifts in temperatures and trends are again related to transitions between regimes. Section 7 presents an analysis of temperature trends in SEA, and for states and regions fully or partially within this area, while Section 8 summarizes corresponding results for Australia as a whole. The implications of our findings and our conclusions are discussed in Section 9. In Appendix A, we consider the relationship between reanalysis data sets and observations in the Australian region, including during the pre-satellite era. The utility and application of decile data, as described in our Methods subsection, is further discussed in Appendix B. Appendix C summarizes regression methods for trends and critical points and their application.

## 2. Material and Methods

### 2.1. Rainfall, Temperature and Streamflow Data Sets

The average and extreme—decile 10—rainfall and temperature data used in this paper have been obtained from the Bureau of Meteorology [61] website. In this study we focus on various regions such as SWWA, SEA, Northern Australia, and Australian states, shown in Figure 1. These Bureau of Meteorology (BoM) data sets are based on averages of station data and are of higher quality than earlier BoM data sets [62,63]. The density of observation sites is particularly high for SWWA and SEA and the starting dates at the beginning of the 20th century have been chosen to ensure high quality [62,63]. For example, Figure 1 of Jones, et al. [63] shows how the number of stations contributing to the rainfall and temperature analyses vary from 1900 and 1910, respectively. Their Figure 2 shows the corresponding networks of stations. The data for streamflow into Perth dams has been obtained from the Water Corporation [64] of Western Australia.

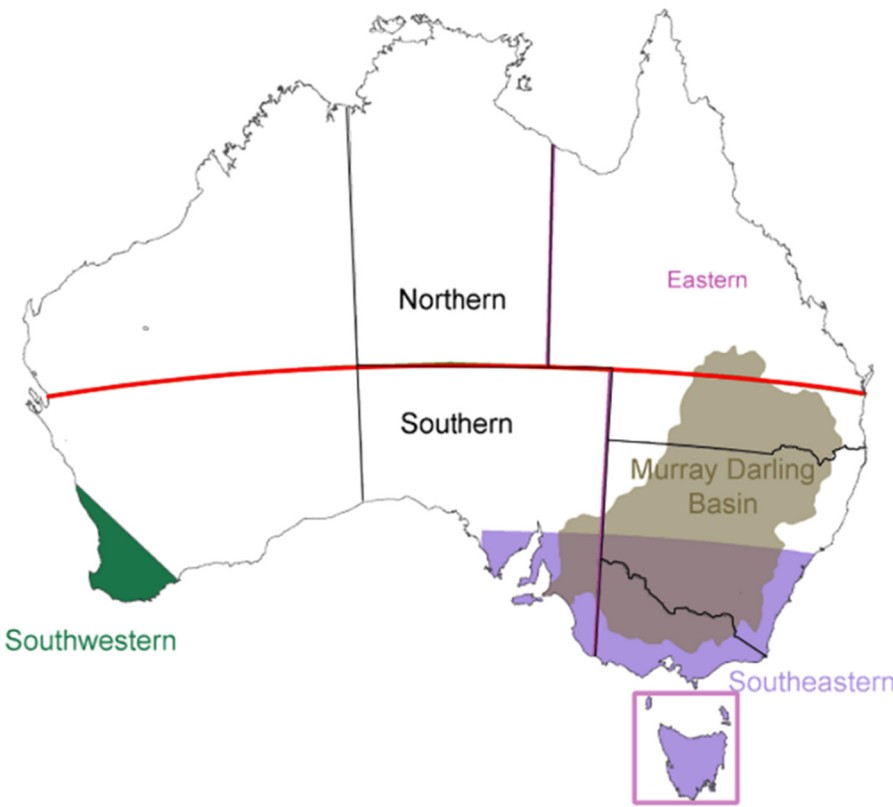

**Figure 1.** Map of Australian regions showing South-western (SWWA), South-eastern (SEA—with a box around Tasmania), Eastern (EA), Southern (SNA), Northern (NA) and Murray Darling Basin (MBD) areas for averaged rainfall and temperature data.

*2.2. Reanalysis Data Sets*

The investigation of the changes in atmospheric circulation in our study uses the reanalysis data set of the National Centers for Environmental Prediction (NCEP) and the National Centre for Atmospheric Research (NCAR), [65]. It will be referred to as the NNR data set. The NNR data set is available for the whole period from 1948 to the present and has been one of the most studied reanalysis data set incorporating the pre-satellite period. We focus here on changes in the Southern Hemisphere zonal jet in the Australian region and their strong relationships to rainfall changes in Southern Australia. In Appendix A, we summarize the results of studies of the comparison of the NNR data set in the Australian region with other reanalysis data sets and with observations.

*2.3. Methods*

Next, we summarize the method of determining decile data from the BoM station data and our method for establishing the changing trends and critical points in data sets.

2.3.1. Determination of Decile Data

Gibbs and Maher [66] developed a decile-based methodology for characterizing meteorological drought and presented Australian maps of the distribution of decile ranges of annual rainfall for the years 1885 to 1965. This method of determining decile data from station rainfall data [63] and temperature data [62] is detailed on the websites of the Bureau of Meteorology [67,68]. Briefly, deciles are a convenient way of coarse-graining the frequency distribution of a variable into ten bands each with 10% of the values. Decile 1 corresponds to the lowest 10%, decile 5 gives the median and decile 10 the highest 10% of the data which is generally monthly, seasonal, multi-month or annual data. The method makes no assumption about the distribution—it is nonparametric—and is based on all the data for a given time span. In practice, for a given time span, gridded data in each grid box,

are sorted from lowest value to highest value and placed into ten equal bands, labelled decile 1 to 10, so that any value in a lower decile is smaller than those in the next decile. The percentage of grid boxes (percentage area) with values in a given decile and year and in a particular geographical region are then calculated based on all the grid boxes, which may be as small as circa 5 km by 5 km (0.05 degrees by 0.05 degrees) for regional rainfall and temperatures [63].

The utility and applications of decile data is further described in Appendix B.

### 2.3.2. Determination of Changes in Trends and Critical Points

The critical times of large and sustained changes in the trends of the rainfall, streamflow and temperature data considered in this study have been determined as follows. The data have been low-pass filtered by applying a 10-year running mean to reduce noise due to the interannual variability. Graphs of the filtered data indicate time periods when these trends change significantly. Regression of the filtered data against time over each time periods incorporating these trend changes are then used to focus in on the critical times. Firstly, regression is applied against a quadratic function of time which highlights the critical time of gradient change. Then, regressions against linear functions of time are performed between the beginning and first critical time, between the last critical time and the end of the timeseries and between any two adjacent intermediary critical points. This then determines the large changes in linear trends we find that are sustained for 15 to 20 years or longer. Averages of the unfiltered data are calculated for the associated time periods.

Appendix C summarizes regression methods for determining trends and critical points and details an example of the application to climate data.

### 3. South-West Western Australian Rainfall, Rainfall Extremes and Atmospheric Circulation

In this section, we analyse rainfall over SWWA and streamflow into Perth dams since the early 1900s. There was a notable deficit in Southern Wet Season (SWS), April to November, rainfall in SWWA and an even larger relative reduction in annual streamflow into Perth dams in the 1960s and 1970s. This has been documented in numerous studies, starting with the articles by Pittock [69] and Sadler, et al. [70], and further analysed and reviewed by Osbrough and Frederiksen [1]. Frederiksen and Frederiksen [6,7] noted that there was an associated 17% reduction in the peak upper troposphere winter jet-stream and a 20% drop in the 300–700 hPa baroclinicity in the region of SWWA between 1949–1968 and 1975–1994. They showed through instability model calculations, with the respective observed climate states for the above two 20-year periods, that there was a circa 30% reduction in the growth rate of leading storm track modes crossing SWWA and a poleward deflection of some storms. Osbrough and Frederiksen [1] have recently confirmed, through a detailed data driven study, that the cause of the SWWA winter rainfall decrease over the last 50 years is in fact the reduction in the intensity of the fast-growing storms associated with changes in the basic state.

Our aim here is to present evidence that both SWS rainfall over SWWA and Perth annual streamflow have undergone regime transitions with qualitative similarities to the phase transitions discussed in the Introduction. Our methodology, as described in Section 2.3.2 is to calculate the critical times of significant changes in the trend of the low-pass filtered mean and decile 10 rainfall and streamflow time series, to calculate these trends, relate them to each other and to changes in the zonal flow around SWWA.

### 3.1. SWWA Rainfall, Rainfall Extremes and Streamflow

We start by examining the time series of SWS rainfall over SWWA and annual streamflow into Perth dams between January to December. The SWWA region is shown in Figure 1 which also displays other regions of Australia that we consider in this study. Figure 2 shows the time series since the early 1900s of SWWA rainfall in SWS, the Percentage Area with Rainfall in Decile 10 ($PAR_{D10}$) for SWWA in SWS and the January to December Perth

streamflow. We note that the three graphs show a general decline with time. This is perhaps most easily seen from Table 1 where averages of these quantities are displayed for different time spans. These time spans have been chosen to capture significant changes in the variables and their trends, discussed in the Methods subsection and below. For each time interval shown the rainfall, streamflow and PAR$_{D10}$ decrease systematically. The reductions shown there are quite profound for streamflow and extreme rainfall. We note that Perth streamflow decreased from an annual average of 414 giga litres for 1911–1958 to 389 giga litres for 1959–1978 to 183 giga litres for 1979–2018 and to as little as 88 giga litres for 2009–2018. Thus, in the last decade Perth streamflow has reduced to just 21% of the historical annual average inflow into dams. Extreme rainfall, represented by PAR$_{D10}$ in Table 1, followed a similar dramatic decrease. By these two measures the climate of SWWA has transited into a completely different regime. Somewhat lesser declines in streamflow have also occurred in other drainage divisions across southern and eastern Australia [2]. For SWWA rainfall (Figure 1a) the broad decreases with time follow a similar pattern to streamflow (Figure 1c) but with the magnitudes of the reductions being considerably less, at circa 20%, since the 1970s.

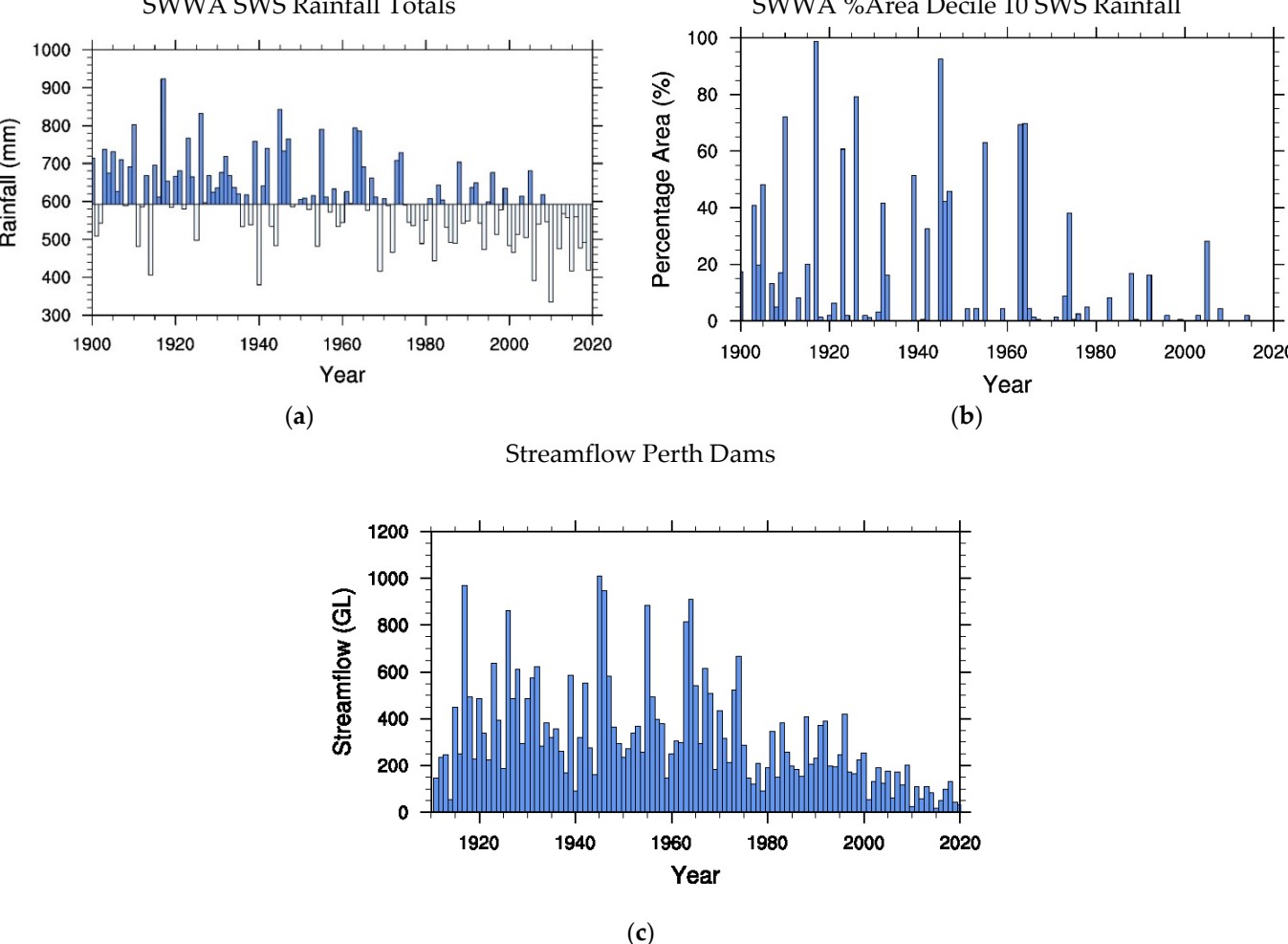

**Figure 2.** Interannual variability of (**a**) 1900–2019 SWWA rainfall totals (mm) for April–November (SWS), (**b**) 1900–2019 percentage area (%) of SWWA with rainfall in decile 10 (PAR$_{D10}$) for SWS and (**c**) 1911–2018 January–December streamflow (GL) into Perth dams.

**Table 1.** The yearly mean SWWA rainfall in SWS, January to December streamflow into Perth dams and percentage area of SWWA with SWS rainfall in decile 10 (PAR$_{D10}$) for different time periods.

| Rainfall and Streamflow | Time Period | | | | |
| --- | --- | --- | --- | --- | --- |
| | **1900–1958** | **1911–1958** | **1959–1978** | **1979–2018** | **2009–2018** |
| SWWA Rainfall SWS (mm) | 639 | | 610 | 545 | 502 |
| Perth Streamflow (gl) | | 414 | 389 | 183 | 88 |
| SWWA % Area Rainfall Decile 10 SWS (%) | 15.5 | | 10.3 | 2.03 | 0.19 |

Table 2 shows correlations (and detrended correlations) for the period 1911–2018 of annual streamflow into Perth dams with SWWA rainfall and PAR$_{D10}$ for SWS. As expected from Figure 2, the correlations are substantial and significant with confidence levels $C_L > 99\%$ in all cases. Correlations are even somewhat larger for the rainfall squared and for a quadratic fit of rainfall with streamflow. Annual streamflow into Perth dams is particularly well described by the quadratic fit with correlation $r = 0.88$ (detrended $r = 0.86$). The fact that streamflow behaves like rainfall squared or through a quadratic fit explains its more dramatic decline.

**Table 2.** Correlations ($r$) between SWWA rainfall, PARD10, streamflow into Perth dams (as in Table 1), rainfall squared and a quadratic fit of rainfall to streamflow with detrended correlations in brackets. The confidence levels ($C_L$) of the correlations are greater than 99% in all cases.

| Correlation Field | SWWA Rainfall SWS | Perth Streamflow Jan–Dec |
| --- | --- | --- |
| SWWA Rainfall SWS | $r = 1.0\ (1.0)$ | $r = 0.84\ (0.81)$ |
| Perth Streamflow Jan–Dec | $r = 0.84\ (0.81)$ | $r = 1.0\ (1.0)$ |
| % Area Rainfall Decile 10 | $r = 0.74\ (0.73)$ | $r = 0.79\ (0.78)$ |
| Rainfall Squared | $r = 0.99\ (0.90)$ | $r = 0.87\ (0.84)$ |
| Quadratic Fit | $r = 0.96\ (0.95)$ | $r = 0.88\ (0.86)$ |

Next, we consider decadal variability of rainfall, streamflow and PAR$_{D10}$. Figure 3 shows time series of 10 year running means of these variables that make the systematic decrease since the mid-1970s more evident than the noisier annual data in Figure 2. The close covariability of the low-pass filtered SWWA rainfall and Perth streamflow is evident and the correlations are even larger ($r = 0.94$ and detrended $r = 0.83$) than for interannual variability ($r = 0.84$ and detrended $r = 0.81$). Perhaps most dramatic is the drop in PAR$_{D10}$ displayed in Figure 3b from before the 1970s to after. This illustrates an important point that how evident a regime transition is depends on the variable of interest and its sensitivity to the changes in the forcings or external environment (order parameters). Clearly extreme rainfall is more sensitive to changes in the circulation that in turn affect the extratropical storms and rainfall [1,7].

The nature of the regime transition can be further elucidated by examining the average trend or gradient of the rainfall and streamflow data over relevant time spans. This is summarized in Table 3 which show the trends up to 1958, between 1959 and 1978 and since 1979. For each of the data sets, there is a considerable decreasing trend in the twenty years between 1959 and 1978 compared with in the periods before and after. Again, these results support the proposition that SWWA rainfall and streamflow into Perth dams underwent a regime transition from a relatively high rainfall state to a lower much drier state and that this occurred over a period of about twenty years. As expected, the broad findings detailed for SWS over SWWA apply equally to winter rainfall and Cool Season (April to October) rainfall (not shown).

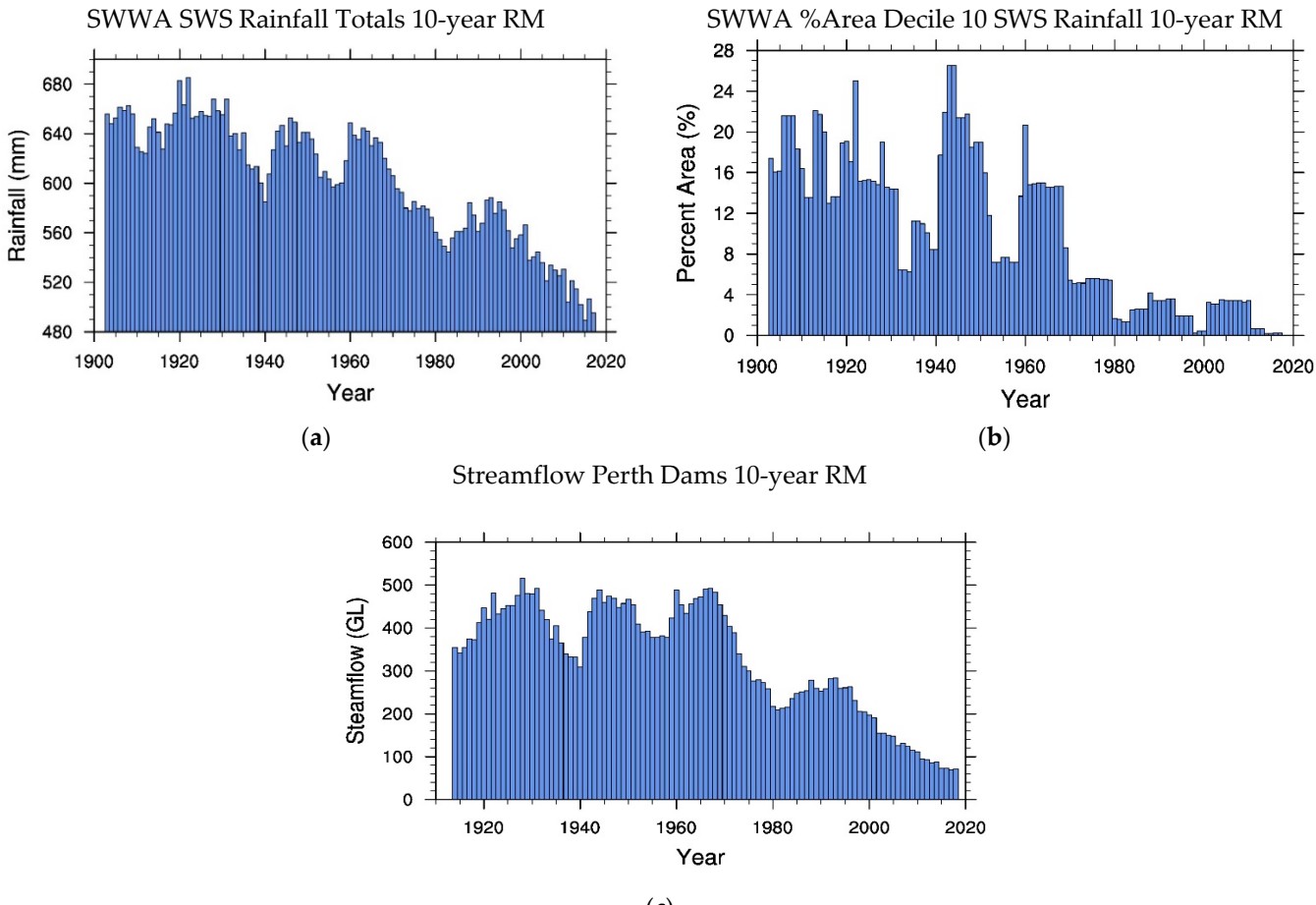

**Figure 3.** As in Figure 2 for 10-year running means (RM) of (**a**) rainfall, (**b**) decile 10 rainfall and (**c**) streamflow data.

**Table 3.** The average gradients or trends of 10-year RM of SWWA rainfall in SWS, January to December streamflow into Perth dams and percentage area of SWWA with SWS rainfall in decile 10 (PAR$_{D10}$) for different time periods.

| | Time Period | | | |
|---|---|---|---|---|
| **Field Gradient** | **1900–1958** | **1911–1958** | **1959–1978** | **1979–2018** |
| SWWA Rainfall SWS 10-year RM (mm year$^{-1}$) | −0.81 | | −4.0 | −1.8 |
| Perth Streamflow 10-year RM (gl year$^{-1}$) | | 0.2 | −11.1 | −5.4 |
| SWWA % Area Rainfall Decile 10 SWS 10-year RM (% year$^{-1}$) | −0.11 | | −0.76 | −0.04 |

### 3.2. SH Atmospheric Circulation

As noted in Frederiksen and Frederiksen [7] and further analysed and reviewed by Osbrough and Frederiksen [1], the July rainfall reduction in SWWA after the 1970s was accompanied by significant decreases in the July upper tropospheric subtropical jet near 30° S over Australia. Here, we examine the time series of the SH jet stream changes since the mid-20th century in more detail focusing on the SWS of April to November. Figure 4 shows a latitude cross section of the (1975–1994) minus (1949–1968) zonal wind difference in the region 90° S–90° N, 100° E–130° E. It has broadly similar structure to the corresponding differences for July shown in Figure 1c of Frederiksen and Frederiksen [7] and Figure 1d of Freitas, et al. [27]. In both cases there are significant wind decreases in the upper troposphere near 35° S, with increases near 60° S and decreases again near 75° S. In Figure 4, the confidence levels $C_L > 95\%$ where the magnitude of the plotted

differences $\geq 2$ ms$^{-1}$; this corresponds closely with the results for July in Figure 1d of [27] that are also statistically significant with $C_L > 95\%$, based on a Student $t$–test. As well these findings are reflected in January to December annual average differences (not shown) indicating the systematic nature of the changes.

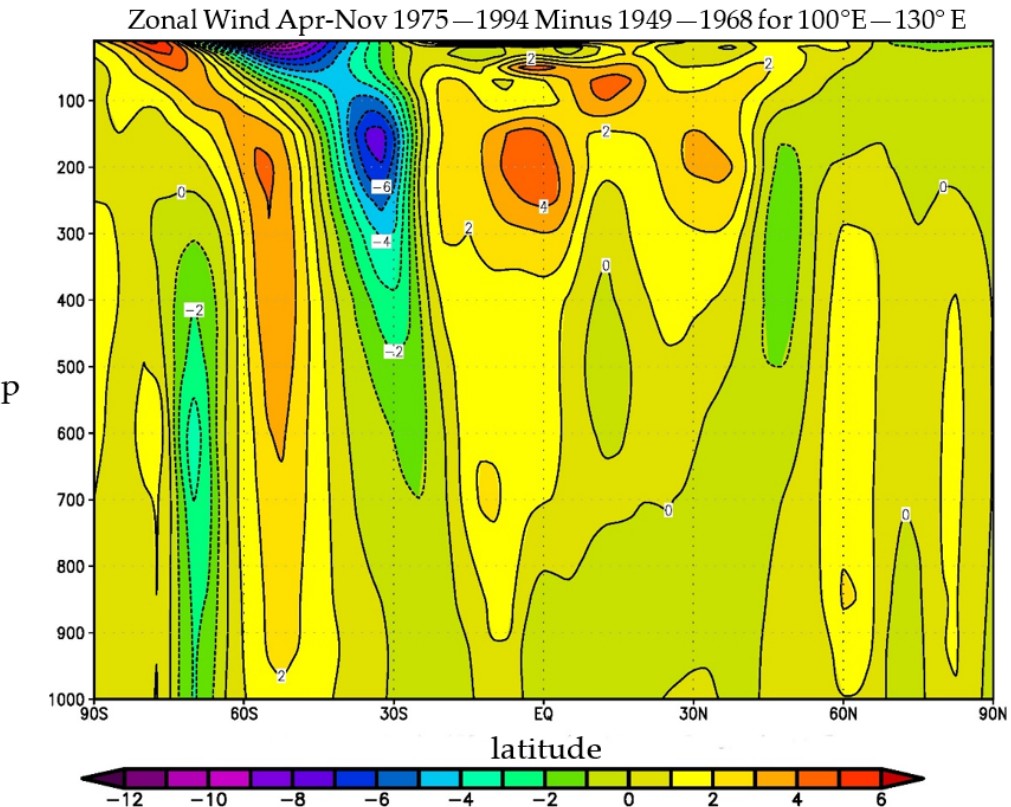

**Figure 4.** Vertical cross-section of April–November zonal wind (m s$^{-1}$) averaged between 100° E–130° E as a function of latitude and pressure (p in hPa) for (1975–1994) minus (1949–1968). Contour intervals are 1 ms$^{-1}$ and confidence levels $C_L > 95\%$ where the magnitude of the plotted differences $\geq 2$ ms$^{-1}$.

Heavy rainfall from rapid extratropical storm development ([1] and references therein) depends on the baroclinicity of the atmosphere. Phillips [71] formulated a simple instability criterion for storm development, based on the zonal wind $U$, which may be expressed as

$$U^{upper} - U^{lower} - U^{critical} > 0$$

which is necessary for baroclinic instability. The superscripts denote the winds at appropriate upper and lower levels of the atmosphere and the critical value, $U^{critical}$, depends on the vertical temperature gradient and the Coriolis parameter. In spherical geometry the expression for $U^{critical}$ is given, for example, by Frederiksen ([72], Equation (3.9)) and in Equation (1) of Osbrough and Frederiksen [1] (and references therein). Frederiksen and Frederiksen [7] and Frederiksen and Frederiksen [73] found that the primary determinant of changes in the SH baroclinicity during the 20th century were changes in the zonal wind shear with changes in the vertical temperature gradient, and thus in $U^{critical}$, being relatively minor. Figure 5 shows the interannual variability of the April to November (SWS) 150 hPa zonal wind (part a) and baroclinicity measured by the 300–700 hPa zonal wind (part b) between 30° S and 35° S, 100° E and 130° E, and from 1948 to 2018 based on NNR data. For both the peak upper tropospheric zonal wind and the baroclinicity there is a general reduction from 1948 until the mid–1970s and thereafter there is a flattening of the running mean curve until the end of the record. These changes are further quantified in Tables 4 and 5. Table 4 show the systematic decrease in 150 hPa zonal wind and 300–700 hPa

tropospheric baroclinicity for the time spans 1948–1958, 1959–1978 and 1979–2018. The corresponding trends or gradients of these field for 1959–1978 and 1979–2018 are given in Table 5. The gradients decrease rather steeply between 1959 and 1978 and thereafter the trend is near zero. In these respects, the results in Table 5 mirror those for SWWA rainfall and stream flow into Perth dams shown in Table 3. The flow field results are consistent with a regime transition into a weaker zonal flow and baroclinicity state in the regions upstream and over SWWA in the twenty-year period 1959–1978. Tables 4 and 5 also show the corresponding changes in the mean values and gradients for the 700 hPa zonal wind in the region 20° S–35° S, 110° E–130° E. We note that in the lower troposphere the relative changes in the mean zonal wind values and particularly gradients are considerably weaker than in the upper troposphere.

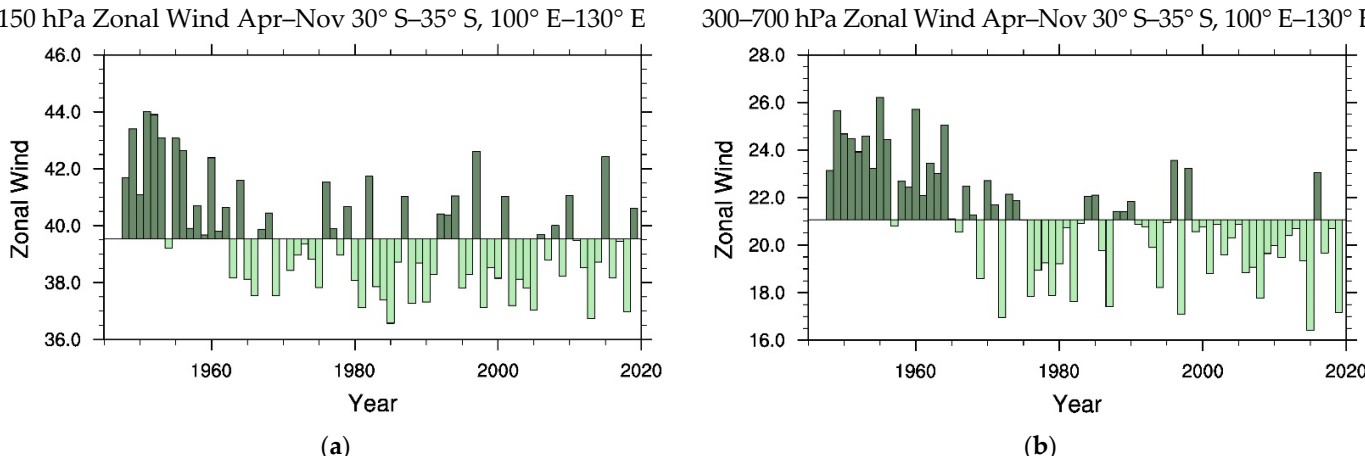

**Figure 5.** Interannual variability of April–November (**a**) 150 hPa zonal wind averaged between 30° S and 35° S, 100° E and 130° E and (**b**) 300 hPa minus 700 hPa zonal wind shear averaged between 30° S and 35° S, 100° E and 130° E.

**Table 4.** The mean April–November (SWS) 150 hPa zonal wind averaged between 30° S and 35° S, 100° E and 130° E, 300 hPa minus 700 hPa zonal wind shear averaged between 30° S and 35° S, 100° E and 130° E, and 700 hPa zonal wind averaged between 20° S and 35° S, 110° E and 132.5° E, for different time periods.

| | Time Period | | |
|---|---|---|---|
| **Mean Zonal Wind** | **1948–1958** | **1959–1978** | **1979–2018** |
| U150 30° S–35° S 100° E–130° E SWS (ms$^{-1}$) | 42.1 | 37.9 | 34.4 |
| U300—U700 30° S–35° S 100° E–130° E SWS (ms$^{-1}$) | 24.0 | 21.4 | 20.1 |
| U700 20° S–35° S 110° E–132.5° E SWS (ms$^{-1}$) | 6.76 | 6.12 | 5.92 |

**Table 5.** The average gradients or trends of SWS 150 hPa zonal wind averaged between 30° S and 35° S, 100° E and 130° E, 300 hPa minus 700 hPa zonal wind shear averaged between 30° S and 35° S, 100° E and 130° E and 700 hPa zonal wind averaged between 20° S and 35° S, 110° E and 132.5° E, for different time periods.

| | Time Period | |
|---|---|---|
| **Mean Zonal Wind Gradient** | **1959–1978** | **1979–2018** |
| U150 30° S–35° S 100° E–130° E SWS 10-year RM (ms$^{-1}$ year$^{-1}$) | −0.33 | −0.003 |
| U300—U700 30° S–35° S 100° E–130° E SWS 10-year RM (ms$^{-1}$ year$^{-1}$) | −0.22 | −0.015 |
| U700 20° S–35° S 110° E–132.5° E SWS 10-year RM (ms$^{-1}$ year$^{-1}$) | −0.018 | −0.019 |

Table 6 shows correlations between mid-tropospheric baroclinicity and characteristics of SWWA rainfall for SWS and annual streamflow into Perth dams for the time span 1959–2018; similar results are obtained for 1948–2018 (not shown). Correlations with the 300–700 hPa zonal wind are as high as 0.58 except with $PAR_{D10}$ where they are lower. We note however that somewhat larger correlations between baroclinicity and rainfall may be obtained by optimizing the region and levels of the flow fields [1]. This is carried out in the right-hand column of Table 6 where correlations (as high as 0.66) with the 700 hPa zonal wind in the region 20° S–35° S, 100° E–132.5° E are shown for SWS. In all cases the correlations are significant with confidence levels $C_L > 98\%$. As noted in [1], the strong correlations of SWWA rainfall with the low-level flow suggests that surface cyclogenesis is a major contributor to the rainfall and the variability of the 700 hPa zonal wind is a primary determinant of variability in low-level baroclinicity. Their results for July are confirmed here for the time span of April to November.

**Table 6.** Correlations ($r$), and detrended correlations in brackets, of SWWA rainfall, $PAR_{D10}$, streamflow into Perth dams, rainfall squared and a quadratic fit of rainfall to streamflow with SWS 300 hPa minus 700 hPa zonal wind shear averaged between 30° S and 35° S, 100° E and 130° E and 700 hPa zonal wind averaged between 20° S and 35° S, 110° E and 132.5° E. Confidence levels $C_L > 99\%$ apart from the detrended correlations with $PAR_{D10}$ for which $C_L > 98\%$.

| Correlation Field | U300–U700 30° S–35° S 100° E–130° E SWS | U700 20° S–35° S 100° E–132.5° E SWS |
|---|---|---|
| SWWA Rainfall SWS | $r = 0.58$ (0.51) | $r = 0.66$ (0.63) |
| Perth Streamflow Jan–Dec | $r = 0.56$ (0.451) | $r = 0.57$ (0.54) |
| Rainfall % Area Decile 10 | $r = 0.36$ (0.29) | $r = 0.34$ (0.29) |
| Rainfall Squared | $r = 0.58$ (0.51) | $r = 0.64$ (0.61) |
| Quadratic Fit | $r = 0.56$ (0.49) | $r = 0.60$ (0.57) |

In summary, our findings in this Section 3 indicate that although the mean and decile 10 rainfall and streamflow since the early 1900s have exhibited interannual and interdecadal variability, the longer time scale variability is characterized by three broad states. During the period up until the end of the 1950s the average trends in these quantities (Table 3) are relatively small compared with the sharp declines in the 1960s and 1970s followed by lesser declining trends since the 1970s. During these three periods there are also systematic decreases in rainfall and streamflow (Table 1) that also have notable correlations with baroclinicity and zonal flows in the region around SWWA (Table 6). It is found that correspondingly upper tropospheric zonal flows and mid tropospheric baroclinicity in the region around SWWA also have significant declining average trends in the 1960s and 1970s with only slight declines after the 1970s (Table 5).

## 4. South-East Australian Rainfall, Rainfall Extremes and Atmospheric Circulation

Next, we examine changes in SEA rainfall since the early 1900s with a particular emphasis on indications of regime transitions as in Section 3 for SWWA rainfall; the aims and approaches are as described there, presented more briefly for the SEA, but also with analysis of variability between individual states within the region. We focus on the Cool Season (CS), April to October, SEA rainfall which is most affected by extra-tropical storms [1]. Perhaps the most dramatic period of rainfall reduction during the 20th and early 21st century was the Australian Millennium Drought (AMD) of 1997 to 2009. SEA rainfall changes, particularly during the AMD, have been the focus of numerous diagnostic studies including by Fawcett [74], Gallant, et al. [75] and Watkins and Trewin [76], and further investigated and reviewed by Osbrough and Frederiksen [1], Dey, et al. [5], Risbey, et al. [11], Cai, et al. [77]. These works have established the AMD as one of the most widespread and devastating droughts of the last century. Frederiksen and Frederiksen [73] related the changes in 1997–2006 rainfall over Southern Australia compared to the 1949–1968 base-line period to changes in the large-scale circulation and changes in the growth of weather

systems. Their theoretical primitive equation calculations showed that the growth rates of leading extra-tropical storm track modes were reduced by more than 30% and onset-of blocking modes by around 20% although there was some increase in the growth rate of North-West Cloud Band modes (NWCBs) and intraseasonal oscillation modes. These theoretical analyses of the causes of the AMD were also supported by the observational study of Risbey, et al. [11], who found fewer fast growing and intense frontal storms and cut-off lows during the AMD and again attributed this to the reduction in baroclinicity in the Australian region. The data driven analysis in Osbrough and Frederiksen [1] confirmed these findings and established that changes in the intensity of explosive storms were primarily responsible for the reduced winter rainfall in Southern Australia during the AMD. They also found that while the El Niños played a significant role in the SEA rainfall reduction during the AMD the general drying of Southern Australia continued and is evident during the longer period 1997–2016.

## 4.1. SEA Rainfall and Streamflow

Figure 6 shows the annual and 10 year running mean time series of SEA rainfall and extreme rainfall characterized by $PAR_{D10}$ for the Cool Season (CS) of April to October; results based on April to November (SWS) are broadly similar (not shown). The reduction in SEA rainfall and $PAR_{D10}$ are most evident from the late 1990s as also see in from Table 7. The SEA reductions in rainfall of about 10% and a halving of $PAR_{D10}$ since the late 1990s are very significant as they affect the Murray Daring Basin (MDB; see Figure 1) which is Australia's main food bowl. Nevertheless, they are not yet as dramatic as the larger reductions experienced by SWWA since the late 1950s discussed in Section 3. As one would expect, many of the states and sub-regions making up, or overlapping with, SEA experienced very similar Cool Season changes as those depicted for SEA. This is the case for the states of Victoria (VIC) and New South Wales (NSW) and for the MDB region (Figure 1). In fact, the variability of rainfall and $PAR_{D10}$ for VIC (the central part of SEA) appears to be synchronous with that for SEA with CS rainfall ($PAR_{D10}$) correlation of 0.97 (0.94). Indeed, the relationships between explosive storms and SEA rainfall established in Osbrough and Frederiksen [1] apply equally to VIC rainfall.

**Table 7.** As in Table 1 for SEA and TAS Cool Season (CS) rainfall and $PAR_{D10}$ and NA Northern Wet Season (NWS) rainfall and $PAR_{D10}$.

| | Time Period | | | | | |
|---|---|---|---|---|---|---|
| **Rainfall** | **1900–1998** | **1999–2019** | **1900–1978** | **1979–2019** | **1900–1968** | **1969–2019** |
| SEA Rainfall CS (mm) | 410 | 361 | | | | |
| SEA % Area Rainfall Decile 10 CS (%) | 11.0 | 5.15 | | | | |
| TAS Rainfall CS (mm) | | | 934 | 914 | | |
| TAS % Area Rainfall Decile 10 CS (%) | | | 12.5 | 5.15 | | |
| NA Rainfall NWS (mm) | | | | | 437 | 509 |
| NA % Area Rainfall Decile 10 NWS (%) | | | | | 5.45 | 16.2 |

The Tasmanian variability of CS rainfall and $PAR_{D10}$ are less representative of SEA with correlations of 0.64 and 0.60, respectively. Interestingly, the changes in TAS rainfall and $PAR_{D10}$ have some similarities to those for SWWA in that the noteworthy reductions in total and extreme CS rainfall commenced in the late 1970s as shown in Figure 7 and in Table 7. However, the Tasmanian rainfall reductions have been more typical of SEA than the larger deficits for SWWA. The reductions in CS rainfall, and extreme rainfall, over the state of South Australia (SA) (not shown) have some similarities with those over SWWA (although not as large) and Tasmania in that they became evident in the late 1970s with further reductions at the start of the 21st century.

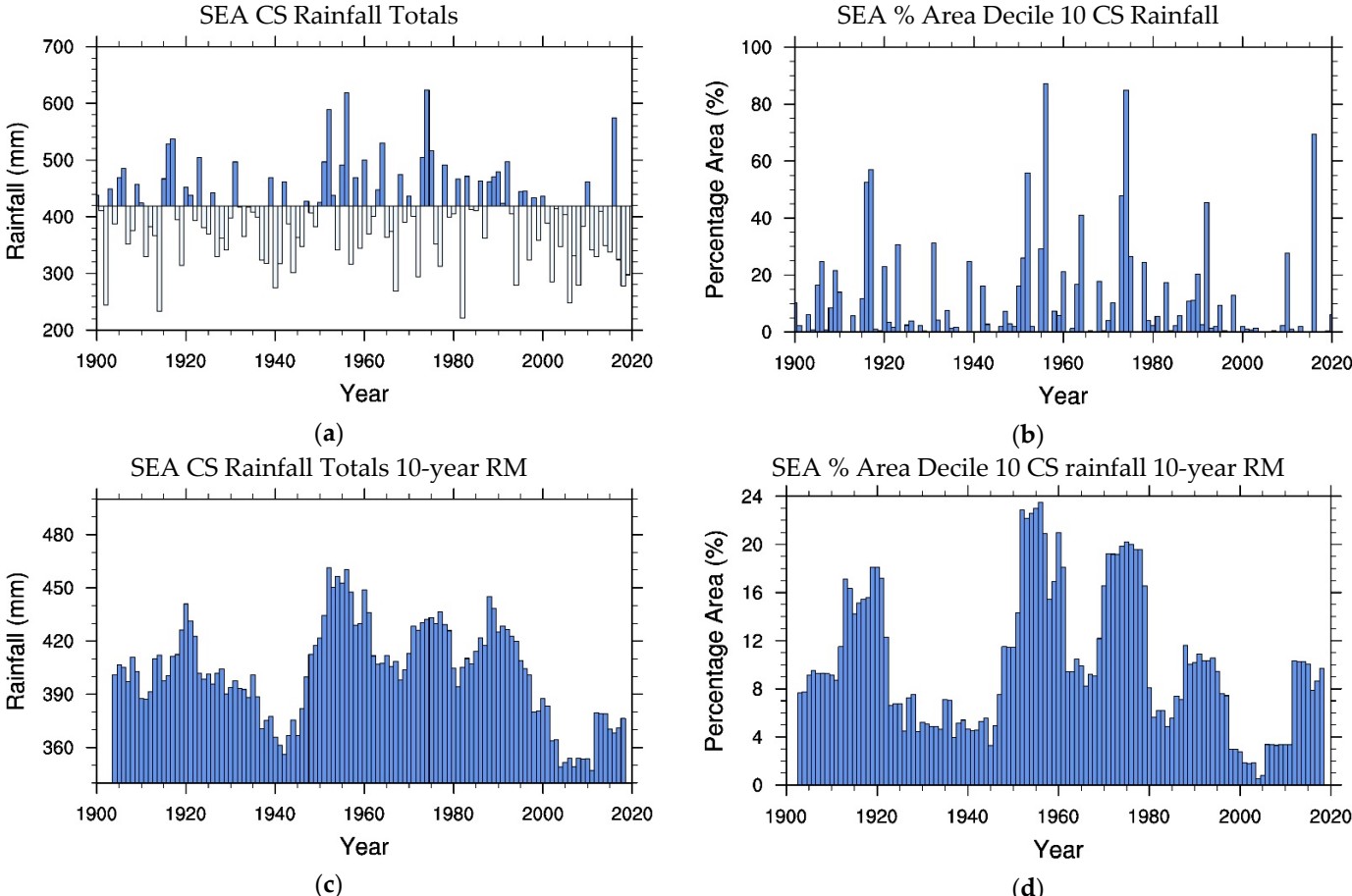

**Figure 6.** (**a**) SEA rainfall totals (mm) for April–October (CS), (**b**) percentage area (%) of SEA with rainfall in decile 10 (PAR$_{D10}$) for CS (**c**) 10-year RM of SEA rainfall totals for CS and (**d**) 10-year RM of SEA PAR$_{D10}$ for CS.

For CS total and decile 10 rainfall averages over the Southern Australian (SNA) and Eastern Australian (EA) regions (Figure 1) the reductions became most evident at the start of the 21st century (not shown); this is also the case for the state of Queensland (QLD) and to a lesser extent even for the Northern Australian (NA) region (not shown).

In this study we shall not make an extensive analysis of the associated changes in streamflow that occurred in SEA or other regions. As might be expected from the relative changes in rainfall between SEA and SWWA the streamflow reductions into some drainage divisions across SEA have been notable but less impactful than those into Perth dams as discussed, for example, in Bureau of Meteorology and CSIRO [2].

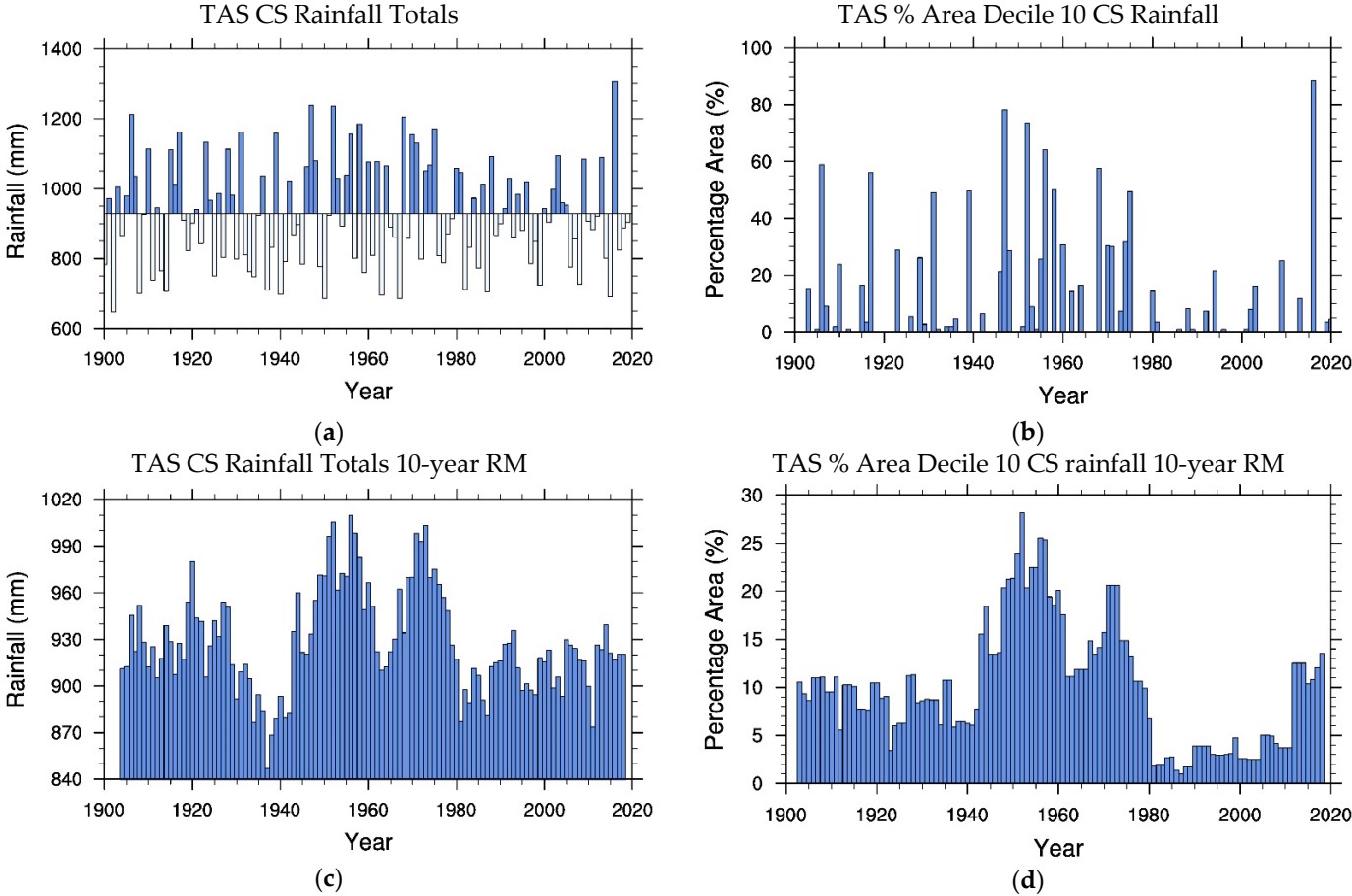

**Figure 7.** As in Figure 6 for TAS (**a**) rainfall, (**b**) decile 10 rainfall, (**c**) 10-year RM rainfall and (**d**) 10-year RM decile 10 rainfall.

### 4.2. SH Atmospheric Circulation

The dynamical study of Frederiksen and Frederiksen [73] noted that the July rainfall reductions during 1997 to 2006, in the AMD, (compared with the baseline 1949–1968 period) were associated with reductions as large as 6 ms$^{-1}$ in the strength of the SH upper tropospheric subtropical jet centred on 30° S between the longitudes of 110° E and 160° E. Similar increases in peak jet strength near 55° S were also noted. From Figure 2 of [73], it is evident that there was also a noteworthy reduction in the baroclinicity of the SH mid-troposphere near 30S particularly in the Australian region. Osbrough and Frederiksen [1] further discussed the changes in the SH circulation, as characterized by several local, hemispheric, and globally important predictors or indices. In particular, they found that July SEA rainfall variability was highly correlated with the 700 hPa zonal wind in the region 20° S–35° S, 132.5° E–155° E. Table 8 shows that on average the correlations between SEA and Tasmanian rainfall and this 700 hPa regional zonal wind are even larger for the seven-month cool season of April to October than for July ([1], Table 4). These correlations are significant with confidence levels $C_L > 99\%$.

In summary, beyond interdecadal variability, SEA has longer time variability characterized by two broad states with the transition between them occurring in the late 1990s (Table 7). This is considerably later than the late 1950s when the first transition of SWWA rainfall occurred as discussed in Section 3. Regime transitions may be triggered by a particular perturbation, or a combination of several, as is the case for extremes such as flooding [16], since it is possible for regime transitions to exhibit hysteresis [44]. In the case of the SEA regime transition we note that the Interdecadal Pacific Oscillation transited from positive to negative phase in the late 1990s ([78], Figure 2) and could be a contributing

trigger that is correlated with SEA rainfall [1]. Interestingly, for Tasmania, making up the southern part of SEA, the transition occurred in the late 1970s between that for SWWA and the central and northern SEA. Again, SEA and Tasmanian rainfall variability is strongly correlated with regional zonal flow variations (Table 8).

**Table 8.** Correlations ($r$), and detrended correlations in brackets, of SEA and TAS Cool Season (CS) and Southern Wet Season (SWS) rainfall with 700 hPa zonal wind averaged between 20° S and 35° S, 132.5° E and 155° E. All confidence levels $C_L > 99\%$.

| Correlation Field | U700 20° S–35° S 132.5° E–155° E |
| --- | --- |
| SEA Rainfall CS | $r = 0.73$ (0.73) |
| SEA Rainfall SWS | $r = 0.72$ (0.73) |
| TAS Rainfall CS | $r = 0.75$ (0.74) |
| TAS Rainfall SWS | $r = 0.73$ (0.73) |

## 5. Northern Australian Rainfall and Rainfall Extremes

While Southern Australia has undergone noteworthy reductions in rainfall since the 1970s, due largely to a reduction in storminess and, particularly, in the intensity of fast-growing extratropical storms [1,6,7,9,11,12,73], Northern Australia (Figure 1) has seen increased precipitation ([2,5,24] and references therein). Table 7 shows the increases in Northern Wet Season (NWS), October to April, total rainfall (of circa 15%) and in extreme precipitation measured by $PAR_{D10}$ (of a nearly three-fold increase) since the late 1960s. These different changes in rainfall in Northern and Southern Australia are to be expected from an expansion of the tropics in the Australian region as noted for example by Lucas, et al. [79] and references therein; their Tables 1 and 2 show the close correspondence between the 1979 to 2001 trends of their measure of tropical expansion from radiosondes and the NNR data in the Australian-New Zealand region between 300 and 100 hPa. The available radiosonde stations for the SH, including for Australia, is shown in Figure 1 of Lucas, et al. [79]. While these increases in rainfall in Northern Australia are of importance, they cannot make up for the decreases that have occurred in the population centres and food bowls of Southern Australia which are our primary concern in this study.

## 6. South-West Western Australian Temperature and Temperature Extremes

Next, we examine the changes in Australian temperatures that have occurred primarily in the latter part of the 20th century and in two decades of the 21st century. The methodology is again as described in Section 2.3.2 and our aim is again to examine regime transitions. Average Australian temperatures have increased by circa 0.9 °C since 1910 with increases in the temperature extremes [3,62,80]. In this Section we start with an analysis of temperatures and temperature extremes over SWWA. Figure 8 shows time series of annual maximum temperatures and annual Percentage Area with Temperatures in Decile 10 ($PAT_{D10}$) for maximum temperatures between 1911 and 2019 for SWWA. Results are presented for variability on the annual timescale as well as for 10 year running means which again bring out the regime transitions. Despite the interannual variability, maximum temperatures have increased considerably from the early 1990s and extreme maximum temperatures from the start of the 21st century. From Table 9 we see that the increase in maximum temperatures since the early 1990s is circa 0.9 °C while the average area experiencing extreme maximum temperatures has increased from a negligible percentage to 46% of SWWA since the start of the 21st century. The average trends, or gradients, of the 10-year running means of SWWA temperatures, shown in Figure 8c,d are presented in Table 10 for the time spans relevant to the above regimes. We note that the gradient associated with the maximum temperature increases by a factor of nearly 5 between the early and late periods shown while the trend in maximum extreme temperatures ($PAT_{D10}$) changes from negligible (1910–2001) to 4.8% year$^{-1}$ (2002–2019). Indeed, the rate of increase in maximum temperatures and $PAT_{D10}$ for the period 2002–2019 is higher than for any of the other major

geographical regions considered next for which corresponding results are also shown in Table 10.

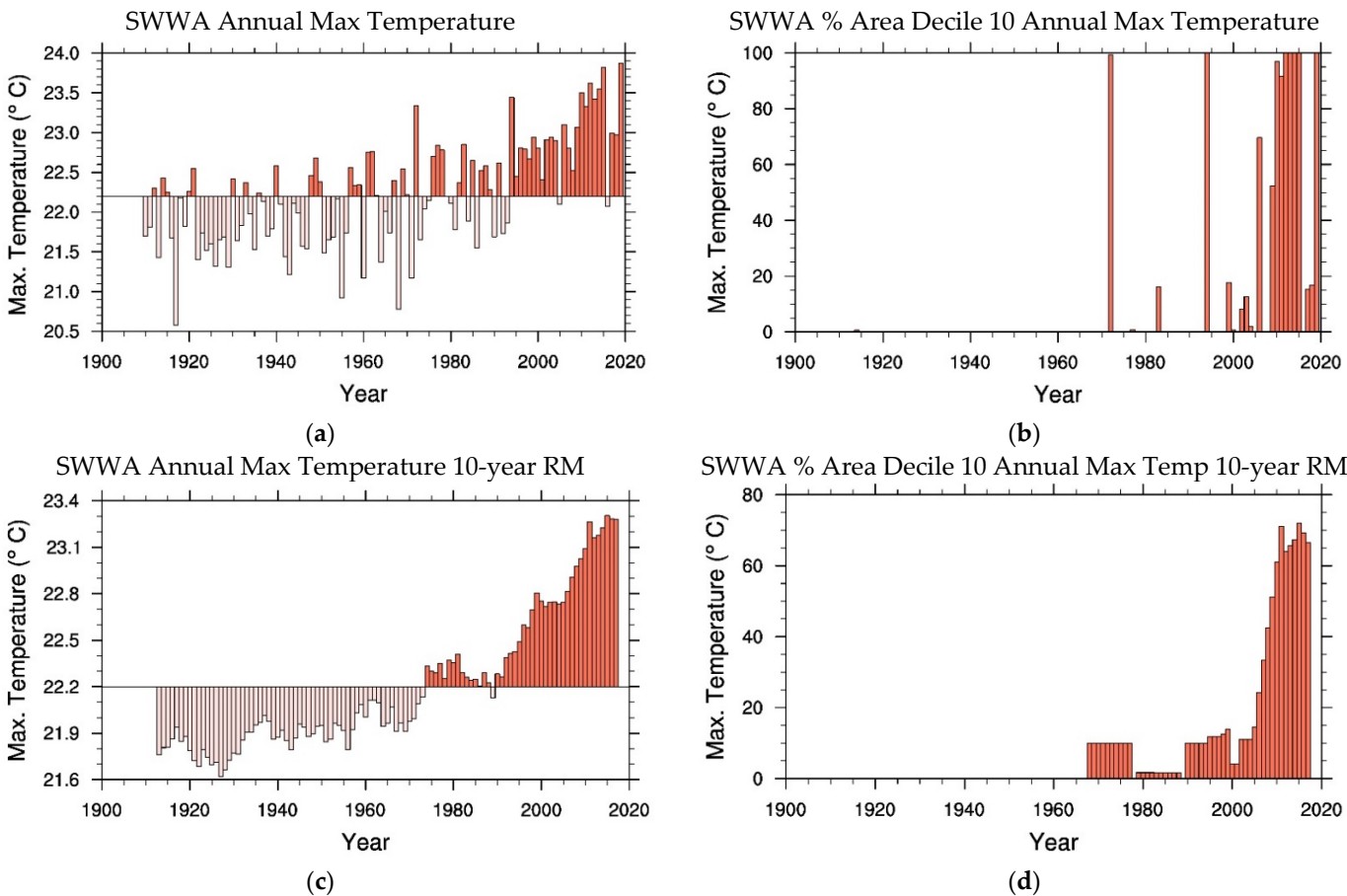

**Figure 8.** (**a**) SWWA annual maximum temperature variability (°C), (**b**) SWWA percentage area (%) with maximum annual temperatures in decile 10 (PAT$_{D10}$), (**c**) 10-year RM of SWWA annual maximum temperatures and (**d**) 10-year RM of SWWA PAT$_{D10}$.

**Table 9.** The annual mean of maximum temperatures and percentage areas with maximum temperatures in decile 10 (PAT$_{D10}$) for SWWA, SEA, NSW, MDB and TAS and Australian PAT$_{D10}$ for mean and maximum temperatures, for different time periods.

|  | Time Period | | | |
|---|---|---|---|---|
| **Mean Field** | **1910–1991** | **1992–2019** | **1910–2001** | **2002–2019** |
| SWWA Max Temp Anomaly Annual (°C) | −0.21 | 0.67 | −0.27 | 0.46 |
| SWWA % Area with Max Temp Decile 10 Annual (%) |  |  | 2.7 | 46 |
| SEA Max Temp Anomaly Annual (°C) |  |  | −0.11 | 1.02 |
| NSW Max Temp Anomaly Annual (°C) |  |  | −0.03 | 1.23 |
| MDB Max Temp Anomaly Annual (°C) |  |  | −0.02 | 1.22 |
| TAS Max Temp Anomaly Annual (°C) |  |  | −0.22 | 0.49 |
| SEA % Area with Max Temp Decile 10 Ann (%) |  |  | 2.7 | 47.2 |
| NSW % Area with Max Temp Decile 10 Ann (%) |  |  | 2.4 | 49.0 |
| MDB % Area with Max Temp Decile 10 Ann (%) |  |  | 2.4 | 49.1 |
| TAS % Area with Max Temp Decile 10 Ann (%) |  |  | 5.7 | 31.9 |
| AUS % Area with Mean Temp Decile 10 Ann (%) |  |  | 3.3 | 44.3 |
| AUS % Area with Max Temp Decile 10 Ann (%) |  |  | 2.8 | 46.6 |

**Table 10.** As in Table 9 for the gradients or trends of 10-year RM of the temperatures and $PAT_{D10}$.

| | Time Period | | | |
|---|---|---|---|---|
| **Field Gradient** | **1910–1991** | **1992–2019** | **1910–2001** | **2002–2019** |
| SWWA Max Temp Anomaly Annual 10-year RM ($^\circ$C year$^{-1}$) | $0.73 \times 10^{-2}$ | $3.7 \times 10^{-2}$ | $0.93 \times 10^{-2}$ | $4.5 \times 10^{-2}$ |
| SWWA % Area with Max Temp Decile 10 Ann 10-year RM (% year$^{-1}$) | | | 0.11 | 4.8 |
| SEA Max Temp Anomaly Annual 10-year RM ($^\circ$C year$^{-1}$) | | | $0.45 \times 10^{-2}$ | $3.0 \times 10^{-2}$ |
| NSW Max Temp Anomaly Annual 10-year RM ($^\circ$C year$^{-1}$) | | | $0.36 \times 10^{-2}$ | $3.5 \times 10^{-2}$ |
| MDB Max Temp Anomaly Annual 10-year RM ($^\circ$C year$^{-1}$) | | | $0.33 \times 10^{-2}$ | $3.2 \times 10^{-2}$ |
| TAS Max Temp Anomaly Annual 10-year RM ($^\circ$C year$^{-1}$) | | | $0.73 \times 10^{-2}$ | $1.9 \times 10^{-2}$ |
| SEA % Area with Max Temp Decile 10 Ann 10-year RM (% year$^{-1}$) | | | 0.002 | 2.7 |
| NSW % Area with Max Temp Decile 10 Ann 10-year RM (% year$^{-1}$) | | | 0.0002 | 2.2 |
| MDB % Area with Max Temp Decile 10 Ann 10-year RM (% year$^{-1}$) | | | −0.02 | 2.2 |
| TAS % Area with Max Temp Decile 10 Ann 10-year RM (% year$^{-1}$) | | | 0.02 | 2.0 |
| AUS % Area with Mean Temp Decile 10 Ann 10-year RM (% year$^{-1}$) | | | 0.14 | 2.9 |
| AUS % Area with Max Temp Decile 10 Ann 10-year RM (% year$^{-1}$) | | | 0.05 | 2.5 |

## 7. South-East Australian Temperature and Temperature Extremes

We now turn to regime transitions of south-east Australian temperatures that began near the start of the 21st century with a focus on maximum temperatures including extreme temperatures. Figure 9a shows the annual anomaly in maximum temperatures over SEA with nearly identical results for VIC (correlation of 0.99) and quite similar results for New South Wales (NSW) and the Murray Darling Basin (MDB) (not shown). The MDB is Australia's main food bowl which stretches inland between Victoria through NSW to southern Queensland (Figure 1). We note from Figure 9a that while there is considerable interannual variability in the graph it is evident that maximum temperatures have increased considerably in the early 21st century compared with the 20th century. This change between centuries is more dramatic when considering extreme temperatures. Figure 9b shows time series of the annual Percentage Area with Temperatures in Decile 10 ($PAT_{D10}$) for SEA maximum temperatures with again nearly identical results for VIC (correlation of 0.99). Again, Figure 9c,d show the corresponding 10-year running mean results corresponding to Figure 9a,b, respectively. It is clear from Figure 9 that during the circa 20 years of the early 21st century there were many occasions when extreme maximum temperatures in the decile 10 band covered large areas of SEA compared with earlier. Figure 10 shows corresponding results for changes in annual maximum temperatures over Tasmania. We note that the temperature increases started earlier than shown for the combined temperatures for SEA in Figure 9.

The main deductions that can be made from the results in Figures 9 and 10 are summarised in Table 9. We note that for the whole of the SEA region, and for NSW and MDB (and VIC—not shown), annual maximum temperature anomalies averaged between 1910 and 2001 are quite small while in the eighteen years of 2002–2019 the differences are circa 1.1 $^\circ$C to 1.2 $^\circ$C. For Tasmania the change is less at circa 0.7 $^\circ$C. We also see that extreme maximum temperatures have become more prevalent. For the 20th century $PAT_{D10}$ for maximum temperatures is just a few percent for SEA, NSW and MDB (and VIC—not shown) but for 2002–2019 the area of extreme maximum temperature rises to between 47% and 49% for SEA, NSW, MDB and VIC and to 32% for Tasmania. These temperature changes for SEA are quite dramatic and if continued could have major implications for the primary food bowl of the Murray Darling Basin.

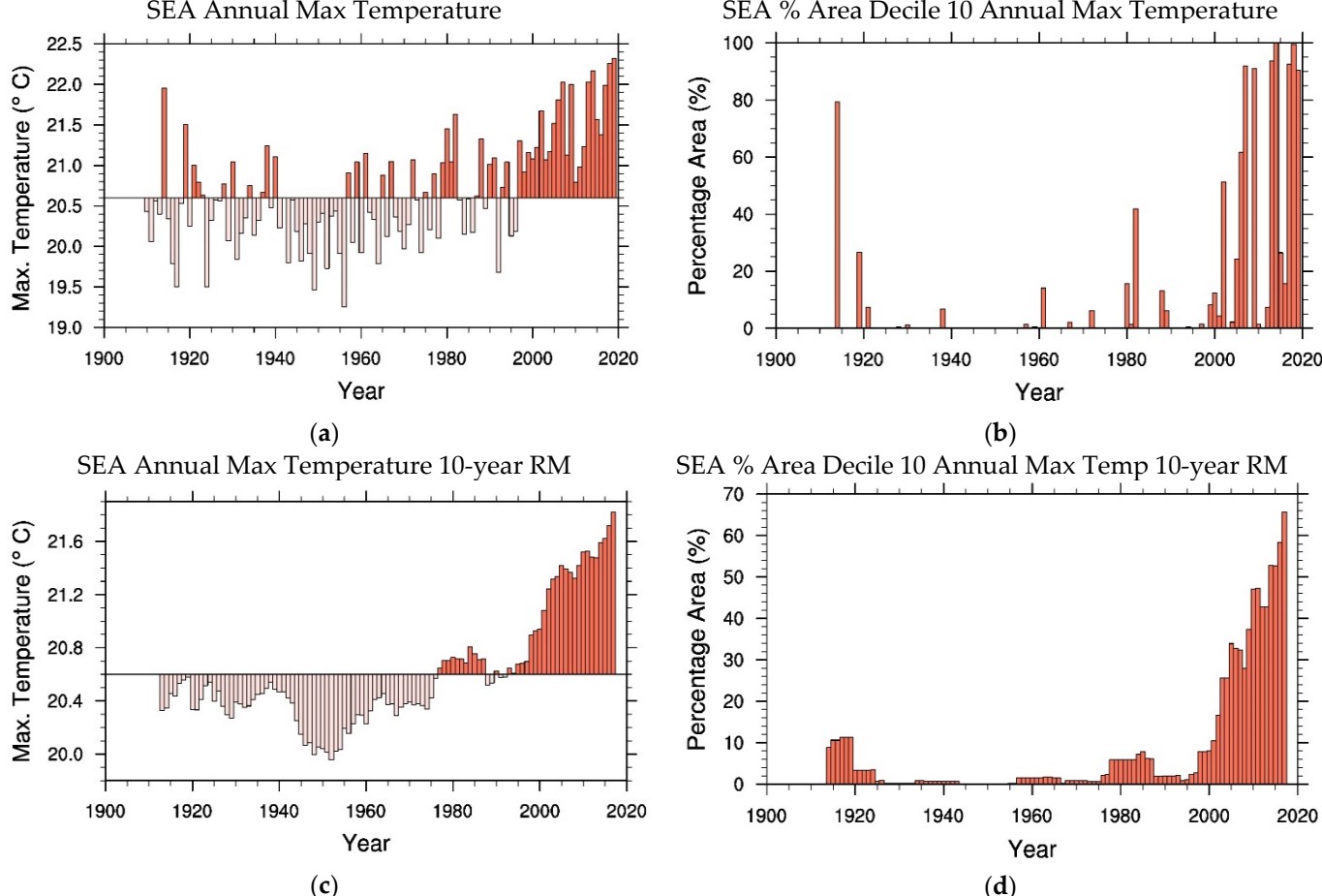

**Figure 9.** As in Figure 8 for SEA (**a**) maximum temperature, (**b**) decile 10 maximum temperature, (**c**) 10-year RM maximum temperature and (**d**) 10-year RM decile 10 maximum temperature.

The smoothed 10-year running means of the results, shown in Figure 9c,d for SEA (respectively, Figure 10c,d for Tasmania), aid in the visualization of the regime transitions. It is evident that maximum temperature anomalies in SEA, and in NSW and MDB, as well as maximum temperature extremes for these regions change dramatically at the start of the 21st century. Table 10 lists the averaged trends or gradients of these 10-year running means of both annual maximum temperature anomalies and temperature extremes for 1910–2001. The distinct increases in trends in the early 21st century again support the concept of a regime transition in SEA temperatures.

For annual average maximum temperatures and $PAT_{D10}$ for maximum temperatures averages over the Southern Australian (SNA) and Eastern Australian (EA) regions (Figure 1), and for the state of QLD, the notable increases again occurred at the start of the 21st century (not shown). For the Northern Australian (NA) region the increases in maximum temperatures and $PAT_{D10}$ started in the 1990s and became more evident during the 21st century (not shown).

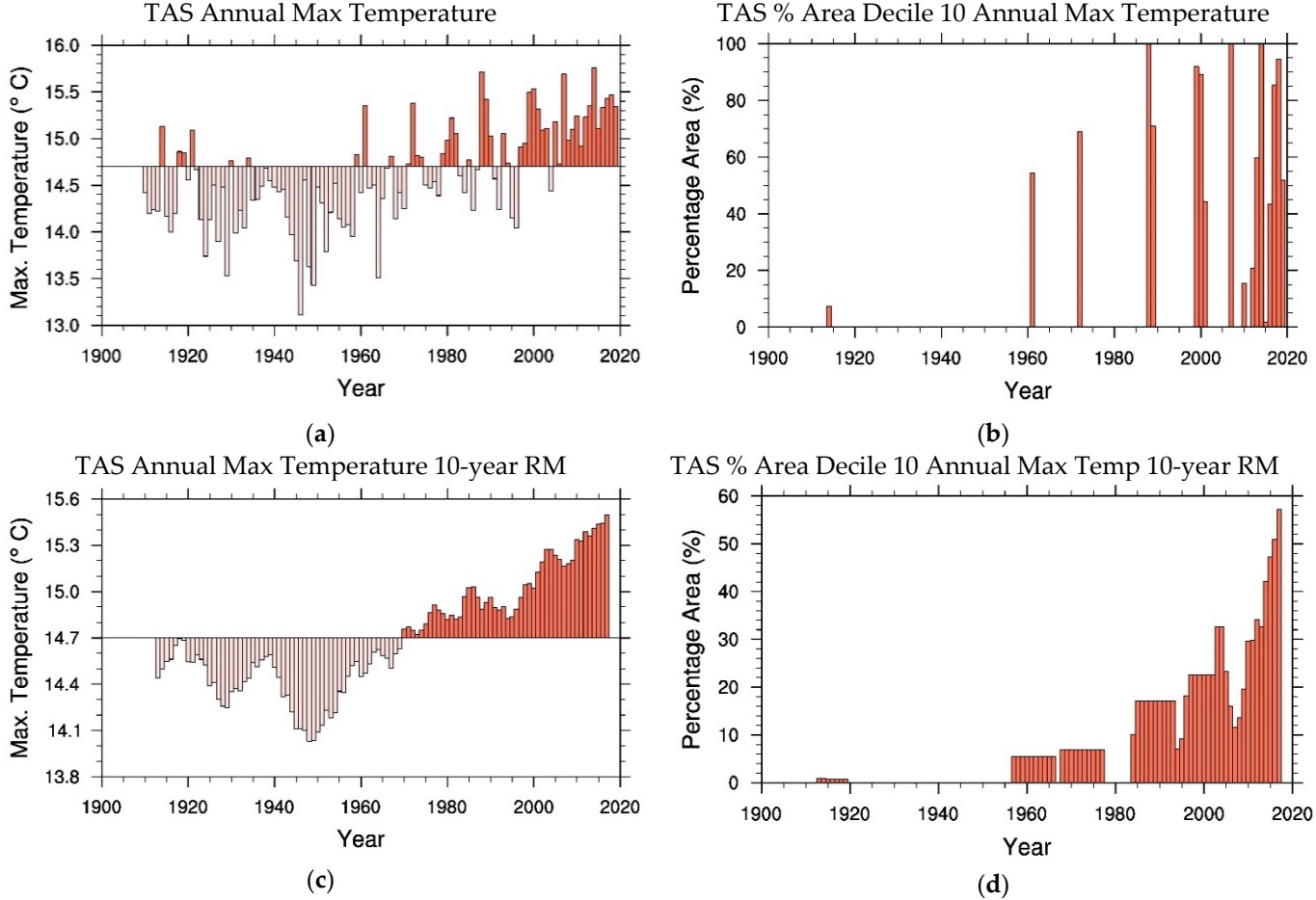

**Figure 10.** As in Figure 8 for TAS (**a**) maximum temperature, (**b**) decile 10 maximum temperature, (**c**) 10-year RM maximum temperature and (**d**) 10-year RM decile 10 maximum temperature.

## 8. Australian Temperature and Temperature Extremes

The regime transitions of SWWA and SEA maximum temperatures and particularly extreme maximum temperatures in fact apply to extreme temperatures averaged more generally across the whole of Australia, as might be expected from the results in the previous sections. For both mean and maximum temperatures extremes characterized by $PAT_{D10}$ increase dramatically in the first two decades of the 21st century; $PAT_{D10}$ increases from just a few percent to 44% for mean and 47% for maximum temperatures for 2002–2019 as show in Table 9. Again, Table 10 shows that there is a considerable change in the averaged gradient around 2002 based on 10-year running means of $PAT_{D10}$ for both mean and maximum temperatures. The distinct increases in trends in the early 21st century again support the concept of a regime transition in Australian temperatures.

In summary, the findings of Sections 6–8 indicate that, for Australia as a whole and for most subregions and states, mean and maximum temperature regime transitions occurred in the early 21st century. In the case of SWWA there are indications that the transitions occurred over the decade of the 1990s.

## 9. Discussion and Conclusions

The main purpose of this study has been to present evidence of regime transitions during the 20th and early 21st century in important aspects of Australian climate. We have focussed on the changes over Southern Australia in rainfall, temperatures and extremes, and associated circulation features since the early 20th century. We have also examined some particularly dramatic shifts in streamflow into Perth dams.

We have found very clear signals that the climate of south-west Western Australia (SWWA) has transited into a drier and warmer state with some of these changes in rainfall, rainfall extremes and streamflow into Perth dams starting as early as the 1960s. Annual streamflow into Perth dams over the last decade has reduced to just 20% of the pre-1960s average. We have determined that the gradient of the 10-year running mean (RM) of streamflow is negligible for the period 1911–1958 followed by a steep decline between 1959 and 1978 and a lesser decline between 1979 and 2018 (circa 40% of that for 1959–1978). Similarly, the Southern Wet Season (SWS) Percentage Area with Rainfall in Decile 10 ($PAR_{D10}$) for SWWA decreased from 16% for the period 1900–1958 to just 0.2% for 2009–2018. As in the case of streamflow into Perth dams most of the decrease in $PAR_{D10}$ occurred between 1959–1978 but with little further systematic decrease between 1979–2018. The changes in magnitude and gradients for streamflow and $PAR_{D10}$ are strong indicators of regime transitions in these variables for SWWA.

Perth streamflow has been shown to be essentially proportional to the square of rainfall and is thus a more sensitive indicator than rainfall itself. Indeed, in comparison SWWA rainfall in the SWS in 2009–2018 was reduced by 21% compared with the long-term average over 1900–1958. Attributing the relative causes of streamflow changes over a large catchment is a difficult problem [81–83]. Liu, et al. [82] in a recent study of streamflow in SWWA catchments concluded that the contributions to streamflow changes were nearly equal for precipitation and catchment characteristics with potential evapotranspiration accounting for a lesser circa 7%. Precipitation can also have an indirect effect in changing catchment properties. This greater sensitivity of streamflow to climate change can also be expressed by the fact that, in broad terms, a 10% reduction in rainfall may reduce streamflow by 30% or more [81,82,84–86].

The gradients for the 10-year RM of rainfall again show similar behaviour to Perth streamflow with the steepest drop between 1959–1978 and a lesser drop between 1959–2018 (circa 45% of that for 1959–1978). As proposed by Frederiksen and Frederiksen [7], and further established in the data driven study of Osbrough and Frederiksen [1] for winter, reductions in the intensity of fast-growing storms in the SWWA region are responsible for the declines in rainfall. We have shown here that the atmospheric flow fields are consistent with a regime transition into a weaker zonal flow and baroclinicity state in the regions upstream and over SWWA in the twenty-year period 1959–1978.

Surface temperatures in SWWA have also increased with most of the rise occurring from the late 20th or early 21st century. Annual maximum temperatures have increased over the last 20 to 30 years by circa 0.9 °C compared with the earlier period starting in 1910. The change in extreme maximum temperatures has been more dramatic; the Percentage Area with Temperatures in Decile 10 ($PAT_{D10}$) increased from 2.7% for 1910–2001 to 46% for 2002–2019. On average, maximum temperatures also increased nearly 5 times faster in 2002–2019 compared with 1910–2001 based changes in the gradient of the 10-year RM. For $PAT_{D10}$ the average gradient was negligible for 1910–2001 but with a rapid rise of 4.8% $year^{-1}$ for 2002–2019. These shifts in maximum temperatures (with somewhat similar shifts in mean and minimum temperatures) and extremes and their gradients are consistent with phase transitions in the temperatures of SWWA in addition to the earlier transitions in rainfall. Interestingly, the alignment of increasing areas of SWWA (Figure 8b,d) to experiencing extreme temperatures since 2002 is reminiscent of the alignment of atomic spins in ferromagnets in transition to the magnetic state.

SWWA has seen the earliest and most dramatic systematic shifts in climate to a drier state with South-Eastern Australia (SEA) impacted towards the end of the 20th century. Cool Season (CS) rainfall over SEA reduced by an average of 12% between the two periods 1900–1998 and 1999–2019 while $PAR_{D10}$ reduced from 11% to 5%. For Victoria (VIC), which is the central region of SEA, the relative changes are virtually identical, and they are also very similar for New South Wales (NSW) and the Murray Darling Basin (MDB). In Tasmania (TAS), the southern part of SEA, rainfall reductions, particular for extreme rainfall, occurred earlier with $PAR_{D10}$ reducing from 12.5% to 5% between the periods

1910–1978 and 1979–2019. Again, the changes in rainfall in SEA is associated with changes in the circulation over and around this region.

Annual maximum temperatures anomalies for SEA, and for VIC, NSW and MDB, averaged over the early period 1910–2001 are quite small while in the eighteen years of 2002–2019 the anomalies are circa 1.0 °C to 1.2 °C. For Tasmania the change is less at circa 0.7 °C. Further, $PAT_{D10}$ for SEA increased from 2.7% for 1910–2001 to 47% for 2002–2019, which is very similar to the case for SWWA and VIC, with slightly larger changes for NSW and MDB, and smaller changes for TAS. For SEA, maximum temperatures rose circa 7 times faster in 2002–2019 compared with 1910–2001 based on changes in the gradient of the 10-year RM. For $PAT_{D10}$ the average gradient was negligible for 1910–2001 but with a rise of 2.7% year$^{-1}$ for 2002–2019.

The regime transitions of SWWA and SEA temperatures are in fact mirrored by shifts over Australia, as a whole. This is seen particularly in extremes, with $PAT_{D10}$ increasing from very low values to 44% for mean and 47% for maximum temperatures for 2002–2019.

We note that there is considerable interannual variability in average and extreme rainfall and temperatures and in streamflow into Perth dams. For that reason, we have also examined 10-year running means to see the systematic changes in the climate variables and their gradients. We have noted discontinuities in the average gradients of the smoothed data typical of second order regime transitions. However, the large and sudden shifts in the temperature extremes are suggestive of first order regime transitions. These abrupt shifts have severe implication particularly for Australian food production, including in the Murray Darling Basin food bowl [2,3], and for increasing frequency and severity of bushfires [2,3,87]. It is aggravated by the continuing downward trend in rainfall, extreme rainfall, and streamflow in Southern Australia [2,3,81,82].

**Author Contributions:** Conceptualization, J.S.F.; methodology, J.S.F.; software, S.L.O.; validation, S.L.O. and J.S.F.; formal analysis, J.S.F. and S.L.O.; investigation, J.S.F.; resources, S.L.O.; data curation, S.L.O.; writing—original draft preparation, J.S.F.; writing—review and editing, J.S.F. and S.L.O.; visualization, S.L.O.; supervision, J.S.F.; project administration, S.L.O.; funding acquisition, S.L.O. All authors have read and agreed to the published version of the manuscript.

**Funding:** S.L.O. was supported through the Climate Systems (CS) Hub of the National Environmental Science Programme (NESP).

**Data Availability Statement:** Atmospheric circulation data is available from the NOAA/ESRL web site: http://www.esrl.noaa.gov/psd/ (accessed on 16 May 2022) and rainfall and temperature data are available from the Australian Bureau of Meteorology web site: http://www.bom.gov.au/climate/change/index.shtml#tabs=Tracker&tracker=timeseries&tQ=graph%3Drain%26area%3Daus%26season%3Dallmonths%26ave_yr%3D0 (accessed on 16 May 2022). The data for streamflow into Perth dams is available from the Western Australian Water Corporation web site: https://www.watercorporation.com.au/Our-water/Rainfall-and-dams/Streamflow (accessed on 16 May 2022).

**Acknowledgments:** We wish to thank David Jones and Blair Trewin of the Australian Bureau of Meteorology for informative discussions on the BoM mean and decile temperature and rainfall data sets. We are grateful for helpful discussions and communications on this work with Yanhui Blockley, Karl Braganza and Branislava Jovanovic of the Australian Bureau of Meteorology, John Ruprecht of Western Land and Water Consulting, Ning Liu of Eastern Forrest Environmental Threat Assessment Center, Artemis Kitsios of Department of Water and Environmental Regulation and Richard Harper of Agricultural Sciences, Murdoch University.

**Conflicts of Interest:** The authors declare no conflict of interest.

## Appendix A. Reanalysis Data Sets and Observations

We have noted the consistency between aspects of the large-scale circulation in the Australian region as characterized by the NCEP-NCAR reanalysis data set [65]; (hereafter NNR) and Southern Australian rainfall in this study and in our earlier works discussed in the Introduction. Kistler, et al. [88] describe the first 50 years of the NNR system and their Figure 1 shows how the number of observations contributing to the analysis has

changed over time at different latitudes. The contributing radiosonde stations, including from Australia, are listed in the National Oceanic and Atmospheric Administration [89] archive and the characteristics of data from 27 Australian radiosonde stations are described by Jovanovic, et al. [90].

In our earlier works (e.g., [12]) we also compared results based on NNR reanalyses with those for European Centre for Medium Range Weather Forecasting (ECMWF) Reanalysis (ERA-40) project [91] and Twentieth Century Reanalysis (20CR v2) project [92] that include the pre-satellite era. Here, we consider the relationships between NNR reanalyses and other reanalyses and observations in the Australian region including during the pre-satellite era.

Hertzog, et al. [93] compared the results of the August 1971 to December 1972 EOLE (from the Greek God of the Winds) experiment, involving flights of super-pressure balloons in the Southern Hemisphere upper troposphere, largely between 230 hPa and 190 hPa and 20° S–70° S, with NNR and ERA-40 data. They argued that their findings are representative of reanalysis accuracy for the pre-satellite era between 1957 and 1979. They noted that their analysis of the zonal wind structure in their Figure 6 shows that both NNR and ERA-40 largely capture the meridional structure of the mean upper tropospheric jet although ERA-40 has a spurious double-jet peak structure. As well, ERA-40 has much larger errors in capturing upper tropospheric synoptic-scale variability. ERA-40 also has larger errors than NNR in representing mean sea level pressure and 500 hPa geopotential heights in the mid to high latitudes of the Southern Hemisphere in the pre-satellite era as shown in Figures 3 and 6 of Bromwich and Fogt [94].

Frederiksen and Frederiksen [6,7], found broad consistency between changes in the Southern Hemisphere July large-scale circulation determined by NNR data, and the results of instability calculations of synoptic disturbances based on this data, and changes in rainfall over south-west Western Australia before and after the mid-1970s. Frederiksen and Frederiksen [7] (Table 7) also compared results for ERA-40 data with those for NNR and found very close agreement for leading synoptic scale modes with growth rates differing by less than 4% and pattern correlations between 0.94 and 0.99. The similar weakening of the mid-winter Southern Hemisphere subtropical jet at 200 hPa around the Australian region, in both NNR and ERA-40 data, in the 1990s compared with the 1950s and 1960s, was also noted by Joseph and Sabin [95] (Figure 4).

Frederiksen, et al. [12], Figures 1 and 2, compared the Southern Hemisphere linear trend in Phillips [71] criterion (discussed in our Section 3.2), in each of the four seasons, for NNR and 20CR v2 over the period 1950–1999 and for ERA-40 over 1958–1999. Of particular interest here are the consistent negative trends in the criterion upstream of Australia in NNR and ERA-40 with generally poorer agreement with 20CR. Rikus [96], in a study of mid-latitude jet streams in nine reanalyses, including NNR, ERA-40 and 20CR, over the period 1979–2009, noted that 20CR had some systematic biases in upper-level winds compared with the other reanalyses. Nevertheless, as shown in Figure 1 of Freitas, et al. [27], the mid-winter reduction in the Southern Hemisphere subtropical jet that occurred in NNR data between the periods (1949–1968) and (1975–1994), and the increase further south, is evident in 20CR v2 data but with peak values at slightly lower levels, consistent with the findings of Rikus [96].

The study of Osbrough and Frederiksen [1], based on six hourly NNR 850 hPa data, found there was good correspondence between the reduction in fast growing storms in the Australian region since the late 1960s and the reduction in Southern Australian rainfall providing further evidence of the general consistency of NNR data with rainfall variability.

## Appendix B. Utility and Applications of Decile Data

Since the first introduction by Gibbs and Maher [66], the decile representation of meteorological data has been widely used in the study of droughts and rainfall and temperature variability. As noted in Section 2, the decile data is available from the website of Bureau of Meteorology [61] and the decile method is described on the websites of Bureau of Meteorology [67,68]. Decile data for seasons, extended seasons and years have been presented in

many reports (e.g., [2]) and BoM publishes Australian maps of annual rainfall deciles from 1900 to the present [97]. Deciles of rainfall and temperatures have also been used in the horticultural and agricultural industries. For example, cool season—April to October—rainfall deciles were used in the South Australian Government study of climate change, wheat production and erosion risk by Sweeney and Liddicoat [98]. In a study of "The Riverland Climate for Almond Production" Thomas and Hayman [99] examined September–April deciles of temperature. Hayman and Hudson [100] explored the value of recent new BoM forecast products of weekly, monthly, and seasonal rainfall and temperature deciles for grain production.

Keyantash [101] reviewed indices of meteorological and hydrological drought and compared the established quantile methodology of deciles with other approaches. He noted the simplicity and nonparametric nature of deciles that make no assumption about the distribution function but determines a coarse-grained version directly from the total available data. Keyantash and Dracup [102] noted that, in the USA context, for Meteorological Drought the rainfall decile index as used at BoM is the superior index overall and particularly in terms of robustness, transparency and extendibility (their Table 3). In the Handbook of Drought Indicators and Indices by the World Meteorological Organization and Global Water Partnership [103] some of the properties of deciles are noted. In particular: deciles are "easy to calculate". "Daily, weekly, monthly, seasonal and annual values can all be considered in the methodology, as it is flexible when current data are compared to the historical record for any given period". "Applications: With the ability to look at different timescales and time steps, deciles can be used in meteorological, agricultural and hydrological drought situations". Table 2 of the Handbook also lists some of the Meteorological Institutions, in addition to those in Australia and USA, that use deciles.

**Appendix C. Regression for Trends and Critical Points**

Here, we summarize and provide an example of the use of regression methods for determining critical points and changes in trends in time series. Suppose we have a time-series $f(t_i)$ for $i = 1, 2, \ldots, N$ and we want to fit it by a polynomial for part or the whole of the timeseries. We shall in fact only need to consider linear and quadratic functions, so we suppose that

$$f(t_i) = a + bt_i + ct_i^2 + \varepsilon(t_i)$$

where $\varepsilon(t_i)$ is the residual in the fit by the quadratic (or parabola). The best fit is achieved if the coefficients are chosen to minimize the sum of the squares

$$S = \sum_{i=1}^{N} \left( f(t_i) - a - bt_i - ct_i^2 \right)^2$$

This minimum is achieved when the derivative of $S$ with respect to each of the parameters vanishes and the optimal parameters are obtained by solving the associated simultaneous equations.

We consider as an example the SEA maximum temperature graph of the 10-year running mean (RM) of the Percentage Area with Temperatures in Decile 10 ($PATD_{10}$) in Figure 8d. We notice that the values are small until after about year 2000 and thereafter $PATD_{10}$ increases rapidly. Suppose we first fit the whole time series with a quadratic. Then, the resultant fit is a parabolic shape that looks like a right-leaning capital J with small values in the early period but too small values near the peak. If we successively shorten the fitted timeseries by starting at later times then the rapidly rising part of the curve becomes better fitted, the correlation with time increases and the parabola becomes more linear. With a start point of 2002 and a linear fit the correlation is 0.95 reflecting the very good fit for the rapidly rising right-hand part $PATD_{10}$. Fitting the left-hand part of the curve through linear regression the correlation with time is 0.02 reflecting a near constant value from the start to 2001. For these fits the root mean square of the sum of the residuals before and after the critical point of 2001–2002 are also minimized. The corresponding consistent

piece-wise trends or gradients are shown in Table 10. Of course, the same results can be arrived at by iterating piece-wise linear regression.

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
