# Peer review of "Tipping Points and Changes in Australian Climate and Extremes"

_climate, doi:10.3390/cli10050073_

Round 1

Reviewer 1 Report

The authors examine systematic changes in temperature and precipitation over Australia covering the period 1960-present. Fairly standard analysis methods (regression, filtering, etc) are applied to decile data. A range of observational data sets are used including streamflow and temperature observations as well as climate reconstructions (reanalysis). The authors findings, that systematic changes in tropospheric dynamics during the late 1970's associated with dramatic reductions in baroclinicity and hence storm activity and cyclogenesis are well supported by the analysis. Seven regions are considered and the respective roles of dynamics (baroclinicity) versus thermodynamics (temperature) determined to account for the observed decline in streamflow. The frequency of occurrence and amplitude of extreme rainfall events are also shown to have declined in many of these regions and is compounded by increased temperatures and evaporation.

Overall, this is a well written, straightforward analysis of data but also supported by the authors substantive body of work on the underlying dynamics at play. That there have been several regime transitions over the past six decades affecting the Australian climate is uncontroversial. The importance of this work is the additional information that regional tipping points have also been exceeded and the quantification of the relative role od climate change.

Reviewer 2 Report

Frederiksen and Osbrough (2022) examine climate regime transitions over Australia by examining rainfall, temperature, and streamflow data in different regions. They found that the southwestern part of Australia has undergone a drying trend and reduced streamflow in recent decades. Since the 1990s, decreases in extreme precipitation have led to such climate regime shifts. Additionally, they found that extreme maximum temperatures have increased in the 21st century. They attribute these shifts to changes in atmospheric circulations. This paper is interesting but rather dense. I wonder if there is a way to synthesize some of these plots and tables as I felt overwhelmed trying to sift through all the plots and connect them with their conclusions. I suggest combining some of the figures and tightening up some of the text. Also, I wonder can the authors attribute these climate regime shifts, particularly the changes in atmospheric circulations, to low-frequency climate variability, such as the Interdecadal Pacific Oscillation (Kiem and Franks 2004), or is it related to climate change, such as global warming? They only mention global warming in the intro.

Lines 28-30 or 356-361: In addition to the Indian Ocean and tropical Pacific, the Atlantic may be related to changes in Australian precipitation. Johnson et al (2018) also examined precipitation variability during the Millennium drought from tropical Pacific and Atlantic forcing through changes in the global Walker circulation. According to their results, a portion of continent-wide Australian precipitation comes from the tropical ocean basins (Ummenhofer et al. 2009).

Line 440: The phase change of the Interdecadal Pacific Oscillation (IPO or PDO) occurred in the late 1990s. Do the authors attribute changes in SEA rainfall to the IPO?

Johnson, Z. F., Chikamoto, Y., Luo, J. J., & Mochizuki, T. (2018). Ocean impacts on Australian interannual to decadal precipitation variability. Climate, 6(3), 61.

Kiem, A. S., & Franks, S. W. (2004). Multi‐decadal variability of drought risk, eastern Australia. Hydrological Processes, 18(11), 2039-2050.

Ummenhofer, C. C., England, M. H., McIntosh, P. C., Meyers, G. A., Pook, M. J., Risbey, J. S., ... & Taschetto, A. S. (2009). What causes southeast Australia's worst droughts?. Geophysical Research Letters, 36(4).

Reviewer 3 Report

Tipping points and changes in Australian climate and extremes

In this study, the authors analyze change points and regime shifts in the Australian climate, streamflow, precipitation, and temperature using 1948-present NCEP/NCAR reanalysis data.

I recommend a major revision, mostly clarification and further explanation of the methodology.

Specific comments

#1 L7: “since the beginning of the 20th century” ... how reliable/rich were the available data at the beginning of the 20th century?  I suggest this should be cautiously discussed in the result/discussion

#2 L78: “approaches unfeasible” I suggest authors mention what kind of feasibility: high computational effort/large uncertainty or data structure/distribution etc.? then how your approach overcomes these limitations. (L76-84) The authors should describe the novelty/significance of their methods more thoroughly/clearly.

#3 Figure 1. rainfall and temperature are averaged over a fairly larger area. Is there any spatial scale that is most suitable to calculate deciles (described in the method section and Table 1)? Should the region be climatologically homogeneous to calculate deciles over the regions? Please describe if any.

#4 section 2.2 Reanalysis data sets: I suggest authors also mention how many observation stations are available across Australia and that are used by NCEP/NCAR. L124-125 unnecessary

#5 section 2.3.2: This is the key to this study. But from the current explanation, it’s difficult to understand the methodology for a non-expert reader.  I suggest the authors explain more clearly/thoroughly. Maybe with an illustration of any particular location.

#6 Discussion section: Stream-flow changes due to watershed disturbance should be critically discussed

 #7 L272: Define U here

#8 L626-627: mention broader implications of the findings of this study in the abstract and conclusion section.

Round 2

Reviewer 3 Report

The authors have responded to all review comments and improved the manuscript accordingly. Now I recommend publication